# Severe COVID-19 Shares a Common Neutrophil Activation Signature with Other Acute Inflammatory States

**DOI:** 10.3390/cells11050847

**Published:** 2022-03-01

**Authors:** Lena F. Schimke, Alexandre H. C. Marques, Gabriela Crispim Baiocchi, Caroline Aliane de Souza Prado, Dennyson Leandro M. Fonseca, Paula Paccielli Freire, Desirée Rodrigues Plaça, Igor Salerno Filgueiras, Ranieri Coelho Salgado, Gabriel Jansen-Marques, Antonio Edson Rocha Oliveira, Jean Pierre Schatzmann Peron, Gustavo Cabral-Miranda, José Alexandre Marzagão Barbuto, Niels Olsen Saraiva Camara, Vera Lúcia Garcia Calich, Hans D. Ochs, Antonio Condino-Neto, Katherine A. Overmyer, Joshua J. Coon, Joseph Balnis, Ariel Jaitovich, Jonas Schulte-Schrepping, Thomas Ulas, Joachim L. Schultze, Helder I. Nakaya, Igor Jurisica, Otávio Cabral-Marques

**Affiliations:** 1Department of Imunology, Institute of Biomedical Sciences, University of São Paulo, São Paulo 05508-000, Brazil; marquesufcg@gmail.com (A.H.C.M.); gabrielacbaiocchi@gmail.com (G.C.B.); freirepp2@gmail.com (P.P.F.); igor.filgueiras@usp.br (I.S.F.); ranieri_twd@hotmail.com (R.C.S.); jeanpierre@usp.br (J.P.S.P.); gcabral.miranda@usp.br (G.C.-M.); jbarbuto@icb.usp.br (J.A.M.B.); niels@icb.usp.br (N.O.S.C.); vlcalich@icb.usp.br (V.L.G.C.); antoniocondino@gmail.com (A.C.-N.); 2Department of Clinical and Toxicological Analyses, School of Pharmaceutical Sciences, University of São Paulo, São Paulo 05508-000, Brazil; carolinealianeprado@gmail.com (C.A.d.S.P.); dennyleandro@gmail.com (D.L.M.F.); desiree.placa@gmail.com (D.R.P.); antedsrocoli@gmail.com (A.E.R.O.); hnakaya@usp.br (H.I.N.); 3Information Systems, School of Arts, Sciences and Humanities, University of Sao Paulo, São Paulo 03828-000, Brazil; gabrieljansenm2000@gmail.com; 4Laboratory of Medical Investigation in Pathogenesis, Targeted Therapy in Onco-Immuno-Hematology (LIM-31), Department of Hematology, Hospital das Clínicas HCFMUSP, Faculdade de Medicina, Universidade de Sao Paulo, Sao Paulo 05403-000, Brazil; 5Department of Pediatrics, Seattle Children’s Research Institute, University of Washington School of Medicine, Seattle, WA 98101, USA; hans.ochs@seattlechildrens.org; 6National Center for Quantitative Biology of Complex Systems, Madison, WI 53562, USA; kovermyer@morgridge.org (K.A.O.); jcoon@chem.wisc.edu (J.J.C.); 7Morgridge Institute for Research, Madison, WI 53562, USA; 8Department of Biomolecular Chemistry, University of Wisconsin, Madison, WI 53506, USA; 9Department of Chemistry, University of Wisconsin, Madison, WI 53506, USA; 10Division of Pulmonary and Critical Care Medicine, Albany Medical Center, Albany, NY 12208, USA; balnisj@amc.edu (J.B.); jaitova@amc.edu (A.J.); 11Department of Molecular and Cellular Physiology, Albany Medical College, Albany, NY 12208, USA; 12Life and Medical Sciences (LIMES) Institute, University of Bonn, 53115 Bonn, Germany; jschrepping@uni-bonn.de (J.S.-S.); j.schultze@uni-bonn.de (J.L.S.); 13Systems Medicine, Deutsches Zentrum für Neurodegenerative Erkrankungen (DZNE), University of Bonn, 53127 Bonn, Germany; thomas.ulas@dzne.de; 14German Center for Neurodegenerative Diseases (DZNE), PRECISE Platform for Genomics and Epigenomics at DZNE, University of Bonn, 53127 Bonn, Germany; 15Hospital Israelita Albert Einstein, São Paulo 05652-900, Brazil; 16Scientific Platform Pasteur, University of São Paulo, São Paulo 05508-020, Brazil; 17Osteoarthritis Research Program, Division of Orthopedic Surgery, Schroeder Arthritis Institute and Data Science Discovery Centre for Chronic Diseases, Krembil Research Institute, University Health Network, Toronto, ON M5T 0S8, Canada; juris@ai.utoronto.ca; 18Departments of Medical Biophysics and Computer Science, Faculty of Dentistry, University of Toronto, Toronto, ON M5G 1L7, Canada; 19Institute of Neuroimmunology, Slovak Academy of Sciences, 845 10 Bratislava, Slovakia; 20Network of Immunity in Infection, Malignancy, Autoimmunity (NIIMA), Universal Scientific Education and Research Network (USERN), São Paulo 05508-000, Brazil

**Keywords:** COVID-19, neutrophil activation, acute inflammatory states, transcriptome profile, integrative analysis of omics data, systems biology

## Abstract

Severe COVID-19 patients present a clinical and laboratory overlap with other hyperinflammatory conditions such as hemophagocytic lymphohistiocytosis (HLH). However, the underlying mechanisms of these conditions remain to be explored. Here, we investigated the transcriptome of 1596 individuals, including patients with COVID-19 in comparison to healthy controls, other acute inflammatory states (HLH, multisystem inflammatory syndrome in children [MIS-C], Kawasaki disease [KD]), and different respiratory infections (seasonal coronavirus, influenza, bacterial pneumonia). We observed that COVID-19 and HLH share immunological pathways (cytokine/chemokine signaling and neutrophil-mediated immune responses), including gene signatures that stratify COVID-19 patients admitted to the intensive care unit (ICU) and COVID-19_nonICU patients. Of note, among the common differentially expressed genes (DEG), there is a cluster of neutrophil-associated genes that reflects a generalized hyperinflammatory state since it is also dysregulated in patients with KD and bacterial pneumonia. These genes are dysregulated at the protein level across several COVID-19 studies and form an interconnected network with differentially expressed plasma proteins that point to neutrophil hyperactivation in COVID-19 patients admitted to the intensive care unit. scRNAseq analysis indicated that these genes are specifically upregulated across different leukocyte populations, including lymphocyte subsets and immature neutrophils. Artificial intelligence modeling confirmed the strong association of these genes with COVID-19 severity. Thus, our work indicates putative therapeutic pathways for intervention.

## 1. Introduction

During almost two years of the COVID-19 pandemic, caused by the severe acute respiratory syndrome Coronavirus (SARS-CoV)-2, over 396 million cases and 5.7 million deaths have been reported worldwide (8 February 2022, WHO COVID-19 dashboard). The clinical presentation ranges from asymptomatic to severe disease manifesting as pneumonia, acute respiratory distress syndrome (ARDS), and a life-threatening hyperinflammatory syndrome associated with excessive cytokine release (hypercytokinemia) [1,2,3]. Risk factors for severe manifestation and higher mortality include old age as well as hypertension, obesity, and diabetes [4]. Currently, COVID-19 continues to spread, new variants of SARS-CoV-2 have been reported and the number of infections resulting in the death of young individuals with no comorbidities has increased the mortality rates among the young population [5,6]. In addition, some novel SARS-CoV-2 variants of concern appear to escape neutralization by vaccine-induced humoral immunity [7]. Thus, there is a need for a better understanding of the immunopathologic mechanisms associated with severe SARS-CoV-2 infection.

Patients with severe COVID-19 have systemic dysregulation of both innate and adaptive immune responses. In addition to highly activated CD4+ T cells [8] and high levels of autoantibodies linked to classic autoimmune diseases [9,10], they present with higher plasma levels of numerous cytokines and chemokines such as granulocyte macrophage colony-stimulating factor (GM-CSF), tumor necrosis factor (TNF), interleukin (IL)-6, soluble IL-6R, IL-8 (CXC chemokine ligand 8 (CXCL8), IL-18, and monocyte chemoattractant protein-1 (MCP-1/CC chemokine ligand 2 [CCL2]) [11,12,13] than patients with moderate or mild COVID-19 disease [14], suggesting a more generalized hyperinflammatory condition. Notably, the hyperinflammation in COVID-19 shares similarities with cytokine storm syndromes such as those triggered by sepsis, autoinflammatory disorders, and metabolic conditions [15,16,17]. For instance, some COVID-19 patients may develop hyperinflammatory conditions such as the multisystem inflammatory syndrome in children (MIS-C), Kawasaki disease, and a severe hyperinflammatory state resembling a hematopathologic condition called hemophagocytic lymphohistiocytosis (HLH) [18]. All of them are life-threatening progressive systemic hyperinflammatory disorders characterized by multi-organ involvement. For instance, HLH patients may develop fever flares, hepatosplenomegaly and cytopenias due to hemophagocytic activity in the bone marrow [18,19,20] or within peripheral lymphoid organs such as the pulmonary lymph nodes and spleen. HLH is marked by the aberrant activation of B and T lymphocytes and monocytes/macrophages, coagulopathy, hypotension, and ARDS.

Therefore, we sought to characterize key signaling pathways and gene signatures associated with this more generalized hyperinflammatory state that is characteristic for some patients with severe COVID-19. The present study represents a follow-up of a recent report from our group [21] in which we performed an integrative analysis of transcriptional alterations in respiratory airways and peripheral blood leukocytes. This approach successfully developed by our and other groups [22,23,24] demonstrated multi-tissue systemic effects of SARS-CoV-2 infection, providing insightful mechanisms of SARS-CoV-2 pathology and cellular targets for therapy [23].

We first compared the molecular overlap between patients with COVID-19 and those with HLH, defined the transcriptomic and proteomic signatures stratifying COVID-19 patients admitted to the intensive care unit (COVID-19_ICU), and then investigated the behavior of the resulting molecular signature in other inflammatory syndromes and infectious diseases, enrolling a total of 1596 individuals whose high throughput data was publicly available (Table 1).

## 2. Materials and Methods

### 2.1. Data Curation 

We searched public functional genomics data repositories (Gene Expression Omnibus [36] and Array Express [37]) for human transcriptome data from patients with HLH and COVID-19 published until February 2021 for our first analyses of common transcriptome signatures between COVID-19 and HLH. During our analyses we included two more recently published COVID-19 datasets, two transcriptome studies from inflammatory diseases and one dataset with cohorts of other respiratory infectious diseases to compare with specific transcriptome signatures resulting from our first analysis. After evaluating the study design, number of samples, and other relevant information (e.g., COVID-19 severity), we obtained raw count files (non-normalized) after trimming and alignment to the reference genome and followed guidelines to perform a meta-analysis report [38], which recommended that we include at least three or four studies to reach a minimum of 1000 participants [39] in order to increase the statistical power of our analysis by increasing the signal-to-noise ratio. This resulted in a cohort of 1596 individuals derived from 11 datasets with transcriptome data generated from different platforms (Table 1).

### 2.2. Differential Expression Analysis and Visualization of Transcriptional Overlap 

Read counts were transformed (log2 count per million or CPM) and differentially expressed genes (DEG) between groups were identified through the webtool NetworkAnalyst 3.0 [40] using the limma-voom pipeline [41]. To determine the DEGs of each dataset we applied the statistical cut-offs of log2 fold-change > 1 (up-regulated), log2 fold-change < −1 (down-regulated), and adjusted *p*-value < 0.05. Shared DEGs among all datasets were displayed using a Venn diagram [42] and Circos Plot [43] online tools. 

### 2.3. Single Cell RNAseq Analysis

The Seurat Object containing the scRNAseq published by Schulte-Schrepping et al. [29] and deposited at the EGA (EGAS00001004571) was used for single cell analysis. We followed the Seurat pipeline [44] as previously described by Stuart et al. [45] to perform differential expression analysis and data visualization, i.e., UMAP, dotplot, and heatmap construction. Regression for the number of UMIs and scaling were performed as previously described [29].

### 2.4. Interactome Analysis

For a more comprehensive Protein–Protein Interaction (PPI) analysis, we used NAViGaTOR 3.0.14 [46] to visualize genes commonly dysregulated in COVID-19 and HLH datasets, highlighting the biological processes enriched by each gene. Prior to visualization, DEGs were used as inputs into the Integrated Interactions Database (IID version 2021-05; http://ophid.utoronto.ca/iid, accessed on 9 February 2022) [47] to identify direct physical protein interactions. The resultant network was then annotated, analyzed, and visualized using NAViGaTOR 3.0.14 [46]. The final figure was combined with legends using Adobe Illustrator ver. 26.0.3.

### 2.5. Enrichment Analysis and Data Visualization

We used the ClusterProfiler [48] R package to obtain dot plots of enriched signaling pathways. Elsevier pathway collection analysis for selected gene lists (seven genes underlying HLH due to inborn errors of immunity (IEI) and 11 genes associated with severe COVID-19) was carried out using the Enrichr webtool [49,50,51]. Sets of genes associated with cytokine/chemotaxis and neutrophil-mediated immunity from each dataset were visualized in bubble-based heat maps applying one minus cosine similarity using Morpheus [52]. Circular heatmaps were generated using R version 4.0.5 (The R Project for Statistical Computing. https://www.r-project.org/ accessed on 4 January 2021) and R studio Version 1.4.1106 (RStudio. https://www.rstudio.com/ accessed on 4 January 2021) using the circlize R package. Box plots were generated using the R packages ggpubr, lemon, and ggplot2. 

### 2.6. Correlation Analysis 

Principal Component Analysis (PCA) of genes associated with COVID-19 severity (25 transcripts) was performed using the R functions prcomp and princomp through the factoextra package [53]. Canonical Correlation Analysis (CCA) [54] of genes associated with cytokines/chemokines and neutrophil-mediated immune responses was performed using the packages CCA and whitening [54]. In addition, we used the corrgram, psych, and inlmisc R packages to build correlograms. Multilinear regression analysis for combinations of different variables (genes) was performed using the R package ggpubr, ggplot2 and ggExtra.

### 2.7. Proteome Data Analysis

We also evaluated the proteomics data obtained from plasma samples of COVID-19 patients previously reported by Overmyer et al. [26]. Briefly, raw LFQ abundance values were quantified, normalized and log2 transformed, as previously described [26]. Differences in protein expression between COVID-19_ICU and COVID-19_nonICU were calculated using the nonparametric MANOVA (multivariate analysis of variance) test [55], followed by the analysis of nonparametric Inference for Multivariate Data [56] using the R packages npmv, nparcomp, and ggplot. Enrichment of differentially expressed proteins (DEP) significant for COVID-19_ICU was performed using the Enrichr webtool [49,50,51] and most significant enriched pathways were displayed by dot plot created with R using tidyverse, viridis and ggplot2 packages, while the Circos Plot of gene-pathway association was built using the Circos online tool [43]. 

### 2.8. Decision-Tree Classification and Machine Learning Model Predictors

We employed a random forest model to construct a classifier able to discriminate between COVID-19_nonICU and COVID-19_ICU, highlighting the most significant predictors for ICU admission. We trained a random forest model using the functionalities of the R package randomForest (version 4.6.14) [57]. Five thousand trees were used, and the number of variables resampled were equal to three. Follow-up analysis used the gini decrease, number of nodes, and mean minimum depth as criteria to determine variable importance. The interaction between pairs of variables was assessed by using minimum depth as the criterion. The adequacy of the random forest model as a classifier was assessed through out of bags (OOB) error rate and the receiver operating characteristic (ROC) curve. For cross-validation, we split the dataset in training and testing samples, using 75% of the observations for training and 25% for testing.

## 3. Results

### 3.1. The Transcriptional Overlap between COVID-19 and HLH

We first performed a cross-tissue analysis of transcriptomic datasets obtained from peripheral blood lymphocytes (PBLs), peripheral blood mononuclear cells (PBMCs), and nasopharyngeal swabs. An association between the transcriptome information across the blood and respiratory airways of COVID-19 patients has been reported by our and other groups [21,23,24]. In this first approach, we obtained a total of 21,583 DEGs from seven COVID-19 cohorts from five datasets (both datasets GSE156063 and EGAS00001004571 have two different cohorts) and one HLH cohort (Figure 1A and Appendix A). Three other COVID-19 studies (GSE163151, GSE152641, and GSE 161731) were only included during our analysis because they were publicly available only after the beginning of our study. To identify the common DEGs we divided the datasets into three subgroups based on the type of samples and RNA seq platforms: Overlap 1 (HLH and COVID-19 blood transcriptomes), Overlap 2 (HLH and COVID-19 nasopharyngeal swab transcriptomes), and Overlap 3 (HLH and COVID-19 scRNA seq transcriptomes) (Appendix A). Even though the total number of DEGs of each dataset has large variability, the number of shared DEGs between the HLH and each COVID-19 dataset was similar across all studies and resulted in a total of 239 unique common DEGs between HLH and all COVID-19 datasets, most of them (237 DEGs) being up-regulated (Figure 1B). Hereafter, we focused on the implications of the up-regulated genes, since the two common down-regulated genes (granulysin or *GNLY;* myomesin 2 or *MYOM2*) alone did not enrich any significant pathway. However, this might also indicate a defect in cytotoxic activity, typical of HLH [58], that will require future investigation. The 237 common up-regulated DEGs encode proteins mainly involved in the immune system, metabolic and signaling processes, forming a highly connected biological network based on physical protein-protein interactions (PPI, Figure 1C). Among them are important molecules involved in the activation of inflammatory immune responses (e.g., PGLYRP1, OLR1, FFAR2), cytokine and chemokine mediated immune pathways (e.g., IL1R2, CXCR2, CXCR8, CCL4, CCL2), and neutrophil activation (e.g., CD177, MPO, ELANE). Of note, the transcriptional overlap between HLH and COVID-19 contains several molecules interacting with 7 HLH/IEI-associated genes, which themselves were not among our DEGs (Figure 1C).

### 3.2. Cytokine/Chemotaxis and Neutrophil Signatures Predominate in COVID-19 and HLH 

We next dissected the biological functions enriched by the 237 common up-regulated DEGs between COVID-19 and HLH patients by performing an enrichment analysis of biological processes (BPs) and cellular components (CCs) by these 237 DEGs. The top 20 most enriched BPs are demonstrated in Figure 2A and encompass cytokine/chemotaxis and neutrophil-mediated innate immune responses ranging from neutrophil activation, degranulation, and migration to responses to IL-1 as well as anti-microbial humoral response, (for all BPs see Appendix A). The CCs enriched (Figure 2B) include several compartments such as the secretory granule lumen and membrane, azurophil tertiary and specific granules, as well as the collagen-containing extracellular matrix, phagocytic vesicles, and primary lysosomes (Appendix A).

Of note, cytokine/chemotaxis and neutrophil signatures predominate in the COVID-19 and HLH multilayered transcriptional overlap. A total of 25, 34, and 58 DEGs are assigned to cytokine, chemotaxis, and neutrophil signatures, respectively (Figure 2C–E: complete categorization can be seen in Appendix A). Several genes play pleiotropic roles in these gene ontology (GO) categories such as *CEACAM8, IL-1β*, *IL-6*, *EDN1*, *NFKB1* and *PDE4B* (Appendix A). For clarity in data visualization, we assigned these genes to a unique category (based on their predominant immunological function according to literature and GeneCards [59] or the human gene database). Among these are genes that code for chemokines and chemokine receptors that attract both lymphocytes and myelocytes to inflammation sites (CCL20, CCL2, CXCR1, CXCR3, CXCL8) [60,61], pro-inflammatory cytokines and cytokine receptors (IL-1B, IL-1R1, NFKB1, IFNG, IL-6, TNF) [62,63] that promote the activation of immune cells, and several proteins/granules with antimicrobial activity (MPO [64], AZU1 [65], ELANE [66,67], DEFA4 [68]). Moreover, there are metalloproteinases (MMP8 and MMP9) involved in the degradation of the extracellular matrix (ECM) to facilitate neutrophil migration [69,70] into the airways and in the regulation of cytokine activity. Of note, hierarchical clustering analysis of these genes indicated a cross-study grouping of closely functional-related molecules. For instance, *IFNG*, *IL6*, and *TNF*; *IL1A* and *IL1B*; as well as genes for signaling molecules involved in the nuclear factor-κB (NF-κB) signaling such as *NFKB1*, *SPHK1*, and *RIPK2* clustered together in the cytokine group. Likewise, *GYPA* and *GYPB (genes encoding blood cell antigens)*, *CEACAM6* and *CEACAM8 (cell adhesion molecules)*, as well as *CCL* and *CXCL,* which encode chemokines in the chemotaxis group, and genes encoding antimicrobial-related peptides such as *AZU1*, *MPO*, *CAMP*, *DEFA4*, *LCN2*, *ELANE*, *OLFM4*, and *CD177* in neutrophil-mediated immunity clustered together. However, we cannot exclude that this clustering pattern just represents a random tendency due to upstream GO categorization.

Relevant literature has emphasized that the seven genes (*AP31B, LYST, PRF1, RAB27A, STX11, STXBP2, UNC13D*) known to cause fHLH (classically defined as familial HLH syndromes and hypopigmentation syndromes) [17,71] contribute to the dominant role played by T and NK cells in the development of HLH [72,73,74]. However, although these 7 genes are not commonly dysregulated across the datasets of COVID-19 and HLH patients, they also enrich several CCs (secretory vesicles, azurophilic granules or specific granules) and BPs (neutrophil degranulation) involved in the neutrophil immune response (Appendix A). This result is in agreement with the role of these genes in a variety of neutrophil functions such as degranulation and formation of neutrophil extracellular traps (NET) [75,76,77,78,79]. 

### 3.3. The Relationship between Cytokine/Chemotaxis and Neutrophil-Mediated Immunity Gene Signatures

Considering the potential association between cytokine/chemotaxis and neutrophil-mediated immunity representing regulatory and effector functions involved in the COVID-19 pathogenesis, we next analysed the relationship pattern and degree between these transcriptional signatures. We chose the COVID-19_PBL dataset (GSE157103) provided by Overmyer et al. [26], which contains transcripts from 100 individuals with COVID-19 and 26 individuals with respiratory symptoms that were negative for SARS-CoV-2, serving as control group (further explored in the next session). We performed canonical-correlation analysis (CCA), which is a multivariate statistical method to determine the linear relationship between two groups of variables [80]. In accordance with the cross-study hierarchical clustering, CCA revealed a strong association between several cytokine/chemotaxis-related genes (e.g., *CXCL8, CEACAMs [1/6/8], IL1RAP, IL1R1, IL1B, NFKB1*) with those involved in neutrophil-mediated immune responses (e.g., *CTSG, ELANE, MMP8, TCN1*) in both COVID-19 patients and controls (Figure 3A,B). Bivariate correlation analysis showed a similar phenomenon (Appendix A). However, these correlation patterns partially changed when comparing COVID-19 with the control group. For instance, while reducing the correlation between DEGs including *IL10, CXCL8, NFKB1, ARG1*, and *SOD2*, new strong associations appeared between *ELANE, DEFA4, AZU1, CTSG*, and *LCN2*, with an overall tendency to higher relationships amid neutrophil-mediated immunity genes in COVID-19 patients. Figure 3C illustrates this observation by scatter plots for some of these variables.

### 3.4. Transcripts Stratifying Severe COVID-19 from Other Respiratory Diseases Are Highly Dysregulated in HLH and Other Acute Inflammatory States

Next, we assessed which genes of cytokine/chemotaxis signaling and neutrophil-mediated immune responses discriminate COVID-19 patients according to disease severity. We further investigated the COVID-19_PBL dataset (GSE157103) [26] comparing COVID-19 patients admitted to the intensive care unit (COVID-19_ICU) with those admitted to non-ICU units (COVID-19_nonICU). The severity of COVID-19 patients at ICU admission was defined based on APACHE II and SOFA scores [81] according to Overmyer et al. [26] (Figure 4A). Among all genes, 25 (15 up-regulated and 10 down-regulated genes) were differentially expressed between COVID-19_ICU and COVID-19_nonICU patients (Appendix A). Of note, most of these 25 genes have also been identified at the protein level as dysregulated in COVID-19 patients across different studies (published during the development of our study; Appendix A). In addition, these 25 genes seem to belong to a systemic immune network of molecules induced by SARS-CoV-2 since they are also highly interconnected with 158 proteins (Appendix A) significantly dysregulated in the plasma of COVID-19_ICU when compared to COVID-19_nonICU patients. Thus, they show several interactions and functional overlap (Figure 4B) with plasma proteins involved in neutrophil degranulation and neutrophil-mediated immunity (Appendix A showing results from protein enrichment analysis). 

Bivariate correlation analysis based on these 25 genes showed that while controls and COVID-19_nonICU patients have a similar general cluster distribution, COVID-19_ICU patients tend to differ, revealing only eight genes with high positive correlations (Figure 4C,D). To investigate the stratification power of these 25 DEGs, we performed principal component analysis (PCA) using a spectral decomposition approach [82,83], which examines the covariances/correlations between variables. This approach revealed that these DEGs clearly divide COVID-19_ICU, COVID-19_nonICU, Control_ICU and Control_nonICU (due to other respiratory illness but negative for SARS-CoV-2) groups (Figure 4E and Appendix A). Likewise, these 25 genes stratified HLH patients from healthy controls (Figure 4F and Appendix A). The PCA indicated that some of these DEGs (e.g., *AZU1,*
*CEACAM8,*
*CTSG, DEFA4, ELANE, LCN2, OLFM4,* and *MMP8*) are more associated with COVID-19_ICU than with COVID-19_nonICU. 

To address whether these 25 DEGs strongly associated with COVID-19_ICU reflect only a specific similarity between COVID-19 and HLH, or if they are also linked to other acute inflammatory states, we investigated the differential expression of these molecules in other inflammatory syndromes and certain infectious diseases. We included additional inflammatory cohorts (GSE178388 [MIS-C] [34] and GSE73461 [KD] [35]) and different respiratory infections (GSE161731 [seasonal coronavirus other than SARS-CoV-2, influenza, bacterial pneumonia] [33]) (Figure 5A). A hierarchical cluster analysis showed the existence of a group of neutrophil-associated DEGs (e.g., *DEFA4, AZU1, ELANE, CTSG, CEACAM8, IL1R1, ARG1,*
*LCN2, OLFM4,*
*MMP8, CD177,* and *MCEMP1*) more consistently up-regulated across all cohorts included in this comparative analysis (Figure 5B and Appendix A). While patients with MIS-C, influenza and seasonal coronavirus showed a similar dysregulation pattern in just a few areas of this cluster of DEGs, patients with KD and with bacterial pneumonia exhibited a similar up-regulation pattern compared to COVID-19_ICU and HLH. Taken together, these data indicate that these DEGs reflect a more generalized inflammatory state rather than being specific to COVID-19 or HLH. 

### 3.5. Multi-Layered Transcriptomic Analysis Associates Neutrophil Activation Signature with COVID-19 Severity

Since scRNA seq allows comparison of the transcriptomes of individual cells, we next sought to investigate the distribution pattern of these 25 genes associated with COVID-19 severity. We analyzed the scRNA seq dataset (EGAS00001004571) reported by Schulte-Schrepping et al. [29] (schematic overview of study group Figure 6A) and found that 21 of the 25 genes associated with COVID-19 severity and HLH development are DEGs among the top 2000 variable genes in the COVID-19 cohort compared to controls (Figure 6B and Appendix A). These 21 genes exhibited cell-type-specific expression patterns. For instance, *CCL4* (a chemoattractant and stimulator of T-cell immune responses [84,85]) was mainly produced by CD8+ T and NK cells, and *CD83* (B, T and dendritic cell activation marker [86,87]) by B cells and monocytes. *CXCL8* was mostly present in monocytes and low-density neutrophils/granulocytes (LDG; also frequently reported as immature neutrophils [88,89,90]), which are neutrophils remaining in the PBMC fraction after density gradient separation. Among these 21 genes, 11 genes (including the eight genes described above), were differentially expressed when comparing patients with mild and severe COVID-19 (Figure 6C). 

Of note, these 11 genes encode proteins that are crucial for several pathways involved in neutrophil-mediated immunity, and are associated with diseases that increase the risk of severe COVID-19 [91,92], such as chronic obstructive pulmonary disease (COPD) [93,94] and ulcerative colitis [95,96] (Appendix A). These 11 genes are also significantly different between COVID-19_ICU and COVID-19_nonICU (Figure 6D) in the bulk-RNA seq dataset (GSE157103, Overmyer et al. 2020 [26]), indicating that these genes are consistently associated with COVID-19 severity across different patient cohorts. Moreover, these 11 genes were differentially expressed in HLH patients compared to healthy controls (Figure 6E). 

We used the random forest method [57] to rank the importance of these 11 genes based on their ability to discriminate between COVID-19_ICU and COVID-19_nonICU in order to evaluate the association of these genes with COVID-19 severity. This approach showed an error rate (out of bag or OOB) of 27,03% and an area under the receiver operating characteristic (ROC) curve of 82,4% for both groups (Figure 7A,B). Follow-up analysis indicated that *ARG1* was the most significant predictor for ICU admission followed by *CD177*, *MCEMP1*, *LCN2*, *AZU1*, *OLFM4*, *MMP8*, *ELANE*, *CTSG*, *DEFA4*, *CEACAM8* based on the number of the nodes, Gini-decrease, and average depth criteria for measuring gene importance (Figure 7C,D). *ARG1* exhibited the most relevant interactions with the other genes according to the mean minimal depth criterion, mostly interacting with *CD177*, *AZU1*, *MCEMP1*, and *LCN2* (Figure 7E). 

## 4. Discussion

Our meta-analysis integrates and unravels the consistency of several important individual studies and datasets that validate the transcriptome data at the protein level in COVID-19 patients [26,29,97]. In agreement with the recent observation that neutrophil hyperactivation plays a key role in the severity of COVID-19 [97,98,99,100], our study indicates that severe COVID-19 disease shares a common neutrophil activation signature with other different acute inflammatory conditions such as HLH [101,102], KD [103,104,105], and bacterial pneumonia [106]. Our data are in agreement with the dual role of neutrophils in providing essential antimicrobial functions, as well as in initiating tissue injury caused by immune dysregulation [107,108]. The genes associated with COVID-19 severity are up-regulated across different leukocyte subpopulations such as lymphoid (NK, T and B cells) and myeloid (monocytes, dendritic cells and LDGs) cells. They form a systemic and interconnected network of cell-type-specific expression patterns and signaling networks that may contribute to the clinical similarities between COVID-19 and other inflammatory conditions. Thus, our analysis identified new candidate biomarkers and novel putative molecular pathways that could lead to novel therapeutic interventions for COVID-19. 

Our work expands the efforts of others [23,109,110,111,112] and of our group [21] to identify networks and pathways involved in the pathogenesis of severe COVID-19. In accordance with our findings, it has recently been demonstrated that neutrophils accumulate in inflamed tissues of COVID-19 patients as a consequence of T-cell driven pro-inflammatory cytokine and chemokine release, which do not return to a homeostatic level due to an ineffective T cell cytotoxic response [101,102]. Moreover, our multi-layered transcriptomics approach is in agreement with the computational model developed by Ding et al. [102], which is based on a network-informed analysis of the interaction of SARS-CoV-2- and HLH-associated genes. This model postulates that neutrophil degranulation/NETs cause endothelial damage and, consequently, thrombotic complications of COVID-19. Ding’s and our interpretation is supported by experimental evidence [97,98,99,113] for neutrophil hyperactivation and its association with the severity of COVID-19, as recently reviewed by Ackermann et al. [100]. As we were able to demonstrate by the multi-omics association between leukocyte and plasma molecules, recently published flow cytometry and proteomic data indicate a systemic and integrated network of molecules associated with neutrophil growth, activation, and mobilization leading to neutrophil dysregulation in severe COVID-19 [98,99]. These results support the concept that the pathophysiology of HLH does not only involve T cell, NK cell and macrophage dysregulation, but also the hyperactivation of neutrophils, as this is also seen in patients with KD [103,104,105] and bacterial pneumonia [106].

Among the common neutrophil activation signatures that is shared by COVID-19 patients and those with other acute inflammatory states, 11 genes (*ARG1, AZU1, CD177, CEACAM8, CTSG, DEFA4, ELANE, LCN2, MCEMP1, MMP8*, and *OLFM4*) commonly dysregulated in COVID-19 and HLH specifically stratified COVID-19_ICU from COVID-19_nonICU patients. They encode proteins involved in neutrophil degranulation and contribute to the development of comorbidities that increase the risk of progressing to severe COVID-19 [91,92]. Random forest model ranking indicated that these genes accurately distinguish COVID-19_nonICU from COVID-19_ICU patients. For instance, this machine learning approach ranked *ARG1* and its interaction with other molecules (*CD177*, *AZU1*, *MCEMP1* and *LCN2*) as an important predictor for ICU admission, supporting the role of these molecules as biomarkers for hyperinflammatory conditions, including those associated with severe COVID-19 [114,115].

Nonetheless, our manuscript has some limitations that need to be considered. The datasets included in our study did not investigate the impact of different SARS-CoV-2 variants on the transcriptome of COVID-19 patients. Hence, further studies are needed to investigate how the different SARS-CoV-2 variants intersect with the other hyperinflammatory conditions that we investigated. We also did not consider the influence of age, sex, and comorbidities on the common transcriptome signatures of COVID-19 and the other hyperinflammatory conditions. In addition, our work requires future mechanistic investigation to further explore and validate the role of molecules studied here as predictors of COVID-19 severity. However, in support of our findings, several of the dysregulated molecules shared by COVID-19 and other acute inflammatory states have been successfully investigated for the treatment of SARS-CoV-2 infection. For instance, inhibition of the CCR5-CCL4 axis by Leronlima (anti-CCR5 monoclonal antibody) [116], or the blockade of cytokine signaling by Tocilizumab (anti-IL-6R) [117], Adalimumab (anti-TNF-α) [118], or Anakinra (anti-IL1R) [119] have been shown to ameliorate, in some cases, severe COVID-19 manifestations. Furthermore, Ruxolitinib, a JAK1/JAK2 inhibitor acting downstream of JAK-dependent chemokines/cytokines such as IFN-γ, IL-1β, IL-6, TNF, G-CSF, CXCL9, and CXCL10 [101,120], has shown promising results in treating COVID-19 [121]. Of note, several approaches targeting neutrophils to treat SARS-CoV-2 complications have entered clinical trials, including the disruption of signaling via CXCR2, IL-8, IL-17A, or the use of phosphodiesterase (PDE) inhibitors [122]. Moreover, in agreement with our data, the inhibition of neutrophil-derived anti-microbial proteins are being actively investigated in clinical trials by exploring the mechanistic and clinical effects of Alvelestat, an oral neutrophil elastase inhibitor (COVID-19 Study of Safety and Tolerability of Alvelestat, ClinicalTrials.gov). In addition, targeting other neutrophil proteins like Azurocidin (*AZU1*) and cathepsin G (*CTSG*) that are elevated in nasopharyngeal swaps of COVID-19 patients, as well as the inhibition of NET formation, have been suggested to alleviate SARS-CoV-2 symptoms [97,101,123].

In conclusion, our comprehensive multi-layered transcriptomic and cross-tissue analysis indicates systemic communalities among severe COVID-19 and other acute inflammatory states. This work suggests an interconnected cytokine/chemokine profile that hyper stimulates and systemically attracts adaptive and innate immune cells, culminating in the hyperactivation of neutrophils. Altogether, these data indicate that both numeric and dysfunctional changes of neutrophils [123,124] are involved in COVID-19 outcomes, i.e., high levels of circulating activated neutrophils [123,125]. Thus, our work suggests common molecular pathways between severe COVID-19 and other acute inflammatory states that can be exploited for therapeutic intervention.

## Figures and Tables

**Figure 1 cells-11-00847-f001:**
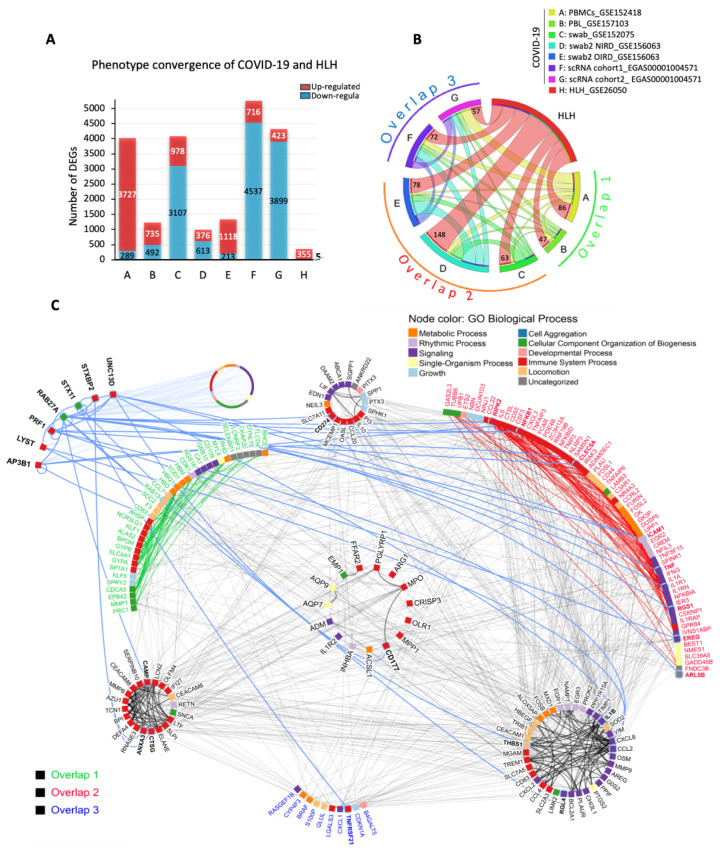
Transcriptional overlap between COVID-19 and HLH. (**A**) Number of differentially expressed genes (DEGs, up- and down-regulated) by dataset. (**B**) Circos plot showing 237 common up-regulated DEGs between HLH and the different COVID-19 datasets (red lines: number at the end of each line indicates exact number of shared DEGs), divided into three overlapping subgroups (detailed in Appendix A). The thickness of each line represents the number of genes shared between the different datasets. (**C**) Protein-protein interaction network among the 237 transcripts and seven genes causing HLH due to inborn errors of immunity (IEI). Node colours denote Gene Ontology Biological Processes. The label (gene name) colours represent transcripts from *Overlap 1* (green), *Overlap 2* (red), and *Overlap 3* (blue). The center circle and side circles represent common molecules across all three or two overlapping datasets, respectively. The upper left subnetwork represents the interactions between the seven genes associated with HLH and those from overlaps are in bold. The circle on upper left (gene names not shown) contains 1329 proteins connected by 217 interactions with the seven HLH/IEI-associated genes. The full network comprises 1538 proteins and 2522 direct physical interactions obtained from the IID database ver. 2021-05 [47].

**Figure 2 cells-11-00847-f002:**
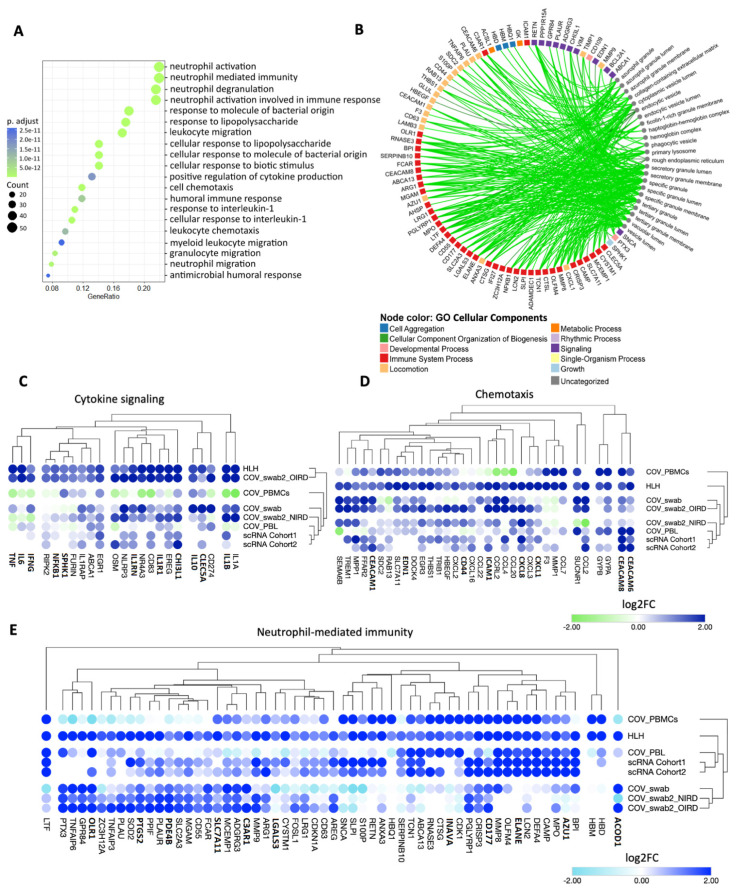
Cytokine/chemotaxis and neutrophil-associated transcriptional signatures predominate in the COVID-19 and HLH overlap. (**A**) Dot plot showing the most significant biological processes (BP) enriched by the 237 common up-regulated transcripts of COVID-19 and HLH datasets. The dot size is proportional to the number of genes enriching the gene ontology (GO) term and color proportional to adjusted *p*-value (green > significant than blue). (**B**) Network highlighting genes and cellular component (CC) associations. Only enriched terms with adjusted *p*-value < 0.05 are shown by small grey circles. The degree of associations is displayed by edge color and thickness (e.g., lighter colors and thinner edges signify fewer connections). Node color represents different GO CCs. Both enriched CCs and BPs were analyzed using ClusterProfiler with R programming. (**C**–**E**) Bubble heatmaps showing the hierarchical clustering based on Euclidian distance of expression patterns of genes associated to (**C**) cytokine signaling, (**D**) chemotaxis, and (**E**) neutrophil-mediated immunity in COVID-19 and HLH datasets. The color of circles corresponds to log2 fold change (log2FC). Pleiotropic genes belonging to more than one category are bold (Appendix A).

**Figure 3 cells-11-00847-f003:**
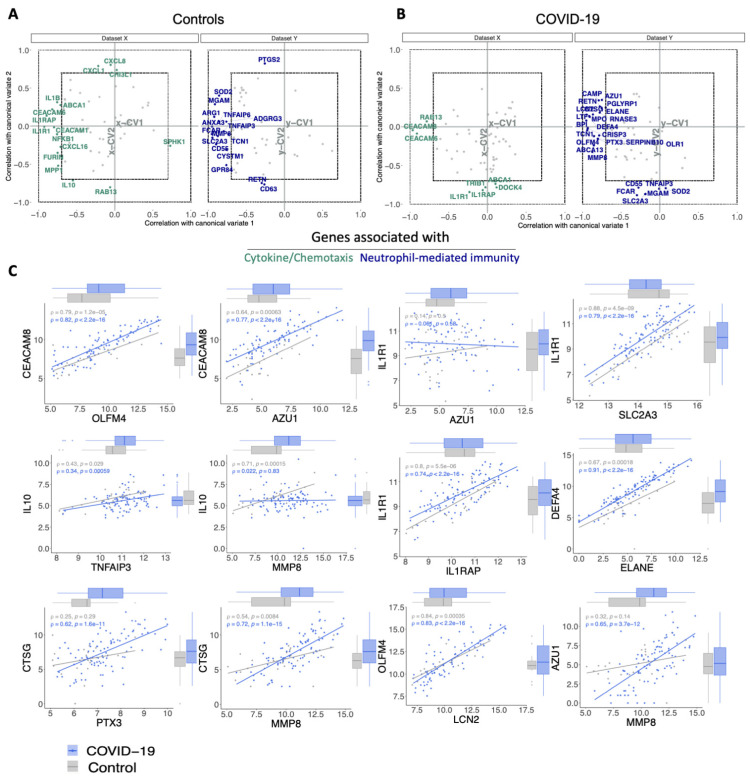
Infection with SARS-CoV-2 impacts the correlation between cytokine/chemotaxis- and neutrophil-mediated immunity genes. (**A**,**B**) Estimated correlations of cytokine signaling/chemotaxis and neutrophil-mediated immunity molecules ranging from −1 to 1 versus their corresponding first two canonical variates (x-CV1 and x-CV2 for cytokine/chemotaxis-related genes; y-CV1 and y-CV2 for neutrophil-mediated immunity genes) in (**A**) controls and (**B**) COVID-19 patients. Cytokine/chemotaxis and neutrophil-mediated immunity genes with a Spearman rank correlation of ≥0.7 are colored in green and blue, respectively, while those with a Spearman rank correlation of <0.7 are gray in both groups. (**C**) Scatter plots with marginal boxplots display the relationship between variables (genes). Correlation coefficient (*ρ*) and significance level (*p*-value) for each correlation are shown within each graph.

**Figure 4 cells-11-00847-f004:**
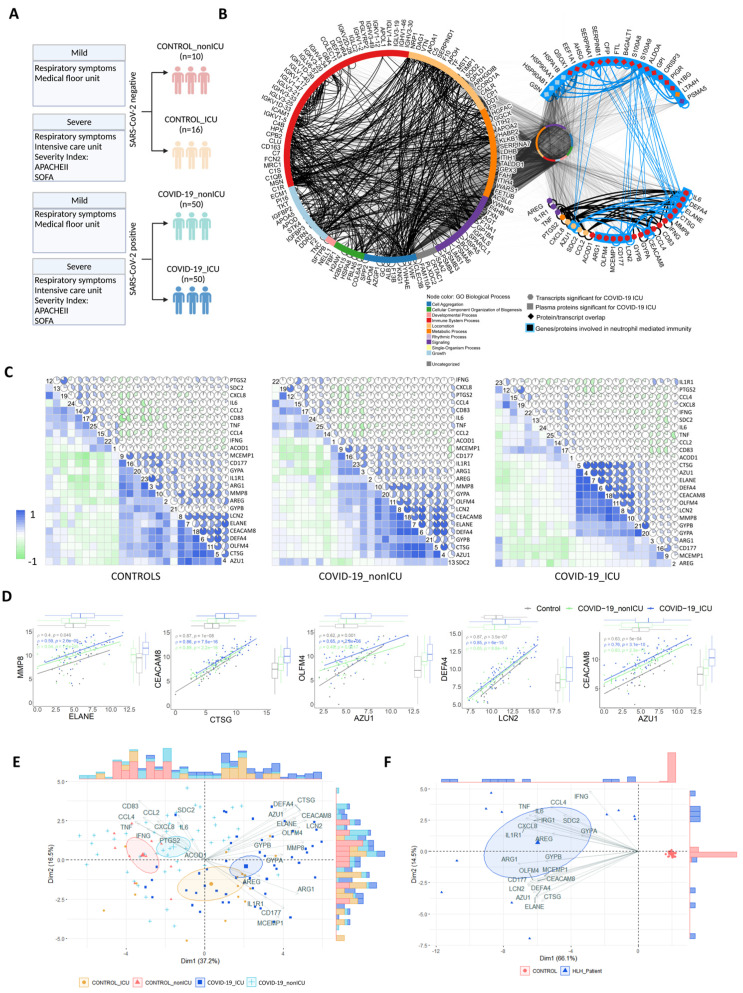
Transcripts stratifying severe COVID-19 from other respiratory diseases and HLH from healthy controls. (**A**) Schematic overview of study design and patient classification of dataset GSE157103 reported by Overmyer et al. [26]. Created with BioRender.com. (**B**) Protein-protein interaction (PPI) network highlighting interactions among the 158 proteins and the 25 genes significant for severe COVID-19_ICU, while keeping their other interacting partners (*n* = 9921) in the middle circle. The node colour denotes Gene Ontology Biological Process terms. The left circle shows 123 proteins and 554 interactions, the upper right half circle shows 21 proteins and 29 interactions, and the lower right side half circle shows 25 proteins and 65 interactions. Molecules involved in neutrophil-mediated immunity are highlighted with a blue node outline. (**C**) Correlation matrices of the 25 DEGs in controls and COVID-19 groups (Controls, left matrix; COVID-19_nonICU, middle matrix; and COVID-19_ICU, right matrix). The color scale bar represents the Pearson’s correlation coefficient, containing negative and positive correlations from −1 to 1, respectively. (**D**) Scatter plots with marginal boxplots display the relationship between the eight genes stratifying severe COVID-19. Correlation coefficient (*ρ*) and significance level (*p*-value) for each correlation is shown within each graph. (**E**) Principal Component Analysis (PCA) with spectral decomposition shows the stratification of COVID-19_ICU from COVID-19_nonICU and other respiratory diseases (Control_nonICU and Control_ICU). Variables with positive correlation are pointing to the same side of the plot, contrasting with negative correlated variables, which point to opposite sides. Confidence ellipses are shown for each group/category. Bar plots associated with the PCA represent the sample distribution across the PCA axes. (**F**) PCA displaying the stratification of HLH patients and healthy controls is based on the same 25 DEGs as in (**E**).

**Figure 5 cells-11-00847-f005:**
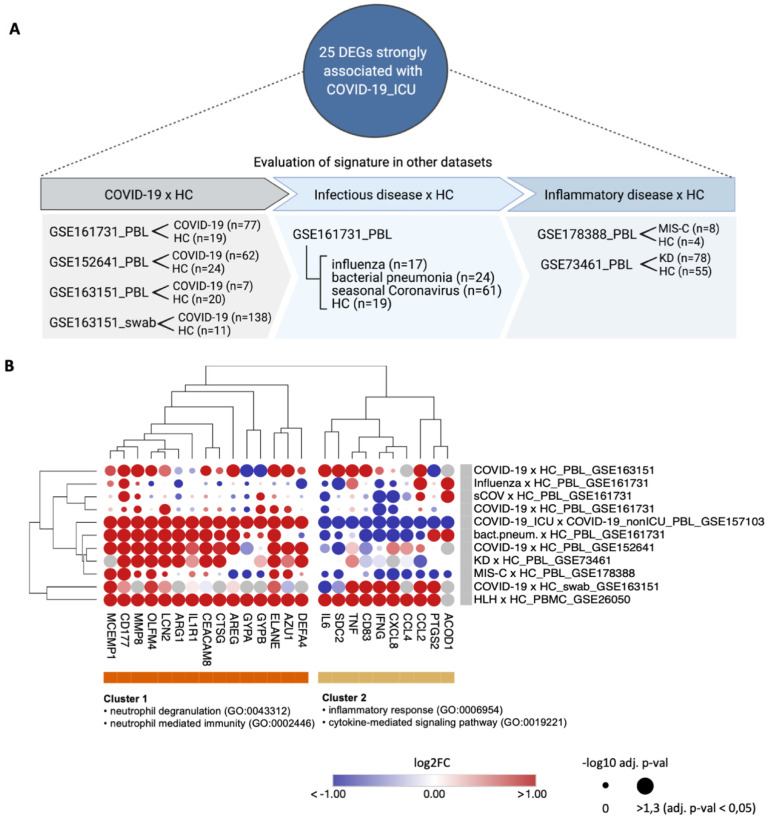
Severe COVID-19 shares a common neutrophil activation signature with other acute inflammatory states. (**A**) Schematic overview of the additional datasets included to evaluate the modulation of the 25 DEGs strongly associated with COVID-19_ICU. Created with BioRender.com. (**B**) Bubble heatmap showing the hierarchical clustering based on one minus spearman rank correlation of the expression pattern of these 25 DEGs across different datasets. Cluster 1 comprises genes associated with neutrophil degranulation and neutrophil-mediated immunity enriched terms, while cluster 2 includes genes enriched in inflammatory response and cytokine-mediated signaling pathway gene ontology (GO) categories. The color of the circles corresponds to the up- and downregulation according to the log2 fold change (log2FC) of each DEG, while the circle size denotes the significant level of each DEG according to the adjusted *p*-value. *HC*, healthy controls; *COV*, seasonal coronavirus other than SARS-CoV-2.

**Figure 6 cells-11-00847-f006:**
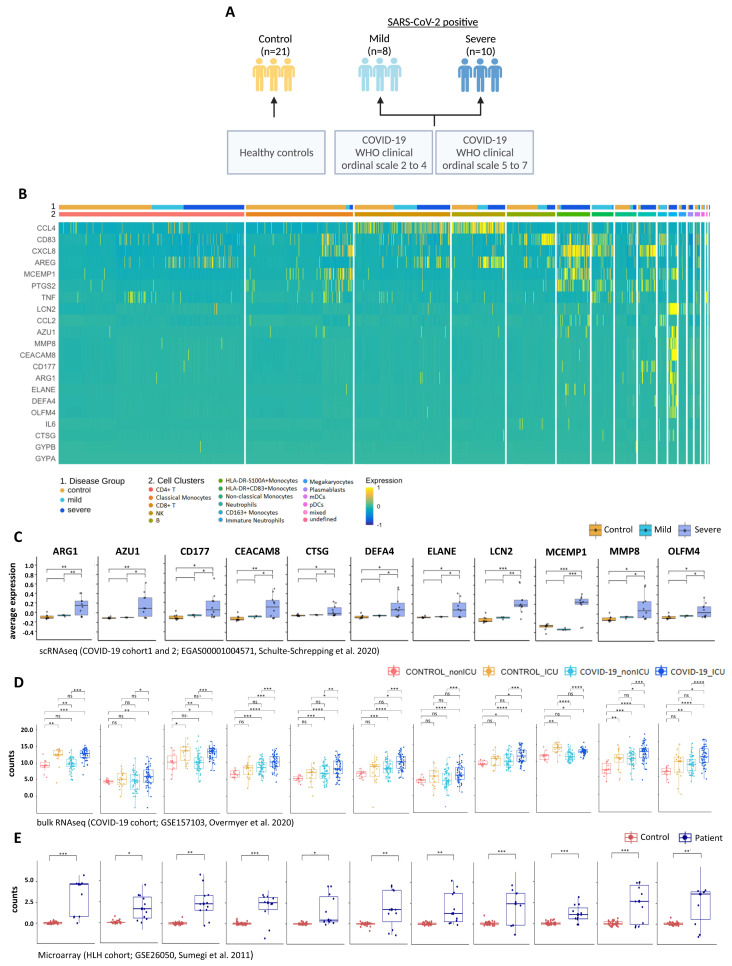
Multi-layered transcriptomic analysis associates neutrophil activation signature with COVID-19 severity. (**A**) Schematic overview of sample cohort and classification of scRNA seq dataset obtained by Schulte-Schrepping et al. [29] and used for the following analysis. Created with BioRender.com. (**B**) Heatmap showing scRNA seq expression of differentially expressed genes (DEGs) associated with disease severity. Cells and cohorts (controls, mild and severe COVID-19) are indicated by different colors in the legends. (**C**) Box plots of scRNA seq expression demonstrating that 11 of the 21 genes identified in (**B**) are up-regulated when comparing severe and mild COVID-19 patients. (**D**) Box plots of the 11 genes stratifying COVID-19_ICU patients from COVID-19_nonICU patients obtained from the bulk-RNA seq dataset from Overmyer et al. [26]. Significant differences between groups are indicated by asterisks (* *p* ≤ 0.05, ** *p* ≤ 0.01, *** *p* ≤ 0.001 and **** *p* < 0.0001). (**E**) Box plots of microarray data illustrating that the disease severity association of COVID-19 detected by scRNA seq corresponds to the expression differences of these genes between HLH patients and controls obtained from the dataset published by Sumegi et al. [30]. Significant differences between groups are indicated by asterisks (* *p* ≤ 0.05, ** *p* ≤ 0.01 and *** *p* ≤ 0.001).

**Figure 7 cells-11-00847-f007:**
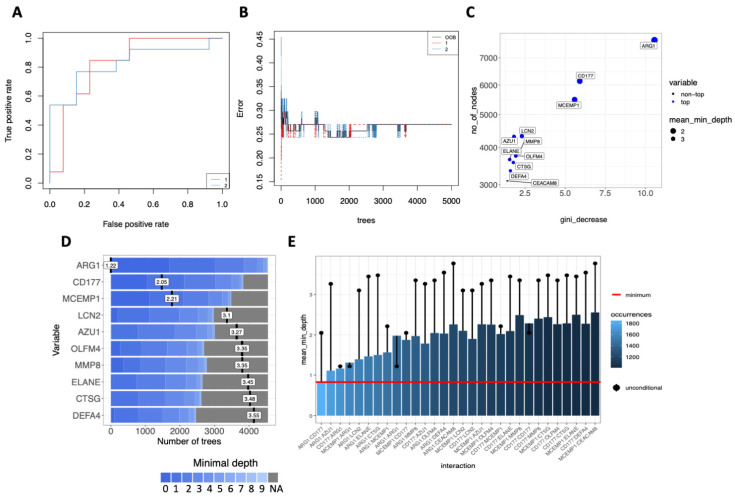
Random Forest prediction analysis suggests potential biomarkers for severe COVID-19. (**A**) Receiver operating characteristics (ROC) curve of 11 genes from COVID-19_ICU compared to COVID-19_nonICU patients with an area under the curve (AUC) of 82.4% for both groups. 1 = COVID-19_nonICU; 2 = COVID-19_ICU. (**B**) Stable curve showing number of trees and error rate (out of bag or OOB) with medium of 27.03%. 1 = COVID-19_nonICU; 2 = COVID-19_ICU. (**C**) Variable importance scores plot based on Gini decrease and number (no) of nodes for each variable showing which variables are more likely to be essential in the random forest’s prediction. (**D**) Ranking of the top 10 variables according to mean minimal depth (vertical bar with the mean value in it) calculated using trees. The blue color gradient reveals the min and max minimal depth for each variable. The range of the *x*-axis is from zero to the maximum number of trees for the feature. (**E**) Mean minimal depth variable interaction plot showing the most frequent occurring interactions between the variables on the left side with light blue color, and least frequent occurring interactions on the right side of the graph with dark blue color. The red horizontal line indicates the smallest mean minimum depth and the black lollipop represents the unconditional mean minimal depth of a variable.

**Table 1 cells-11-00847-t001:** Dataset Information and sample size used for transcriptome analysis.

Data-Base	Dataset ID	Seq. Method	Sample Type	Disease Type of Patients (Sample Size)	Type of Controls (Sample Size)	Original Study
GEO	GSE152418	bulk-RNA seq	PBMC	COVID-19 (*n* = 17)	healthy controls (*n* = 17)	Arunachalam et al., 2020 [25]
GEO	GSE157103	bulk-RNA seq	PBL	COVID-19_ICU (*n* = 50) COVID-19_nonICU (*n* = 50)	SARS-CoV-2 negative ICU (*n* = 16), SARS-CoV-2 negative nonICU (*n* = 10)	Overmyer et al., 2020 [26]
GEO	GSE152075	bulk-RNA seq	nph swab	COVID-19 (*n* = 430)	SARS-CoV-2 negative (*n* = 54)	Liebermann et al., 2020 [27]
GEO	GSE156063	bulk-RNA seq	nph swab	COVID-19 (*n* = 93)	NIRD (*n* = 100) OIRD (*n* = 41)	Mick et al., 2020 [28]
EGA	EGAS 00001004571	scRNA seq	PBL/ PBMC	Cohort1: COVID-19 mild (*n* = 8), COVID-19 severe (*n* = 10) Cohort2: COVID-19 (*n* = 17)	healthy controls (*n* = 21) healthy controls (*n* = 13)	Schulte-Schrepping et al., 2020 [29]
GEO	GSE26050	microarray	PBMC	HLH (*n* = 11)	healthy controls (*n* = 33)	Sumegi et al., 2011 [30]
GEO	GSE163151	bulk-RNA seq	nph swab PBL	COVID-19 (*n* = 138) COVID-19 (*n* = 7)	healthy controls (*n* = 11) healthy controls (*n* = 20)	Ng et al., 2021 [31]
GEO	GSE152641	bulk-RNA seq	PBL	COVID-19 (*n* = 62)	healthy controls (*n* = 24)	Thair et al., 2021 [32]
GEO	GSE161731	bulk-RNA seq	PBL	COVID-19 (*n* = 77) influenza (*n* = 17) bact. pneum. (*n* = 24) seasonal CoV (*n* = 61)	healthy controls (*n* = 19)	McClain et al., 2021 [33]
GEO	GSE178388	bulk-RNA seq	PBL	MIS-C (*n* = 8)	healthy controls (*n* = 4)	Beckmann et al., 2021 [34]
GEO	GSE73461	microarray	PBL	KD (*n* = 78)	healthy controls (*n* = 55)	Wright et al., 2018 [35]

*nph*, nasopharyngeal; *PBMC*, peripheral blood mononuclear cells; *PBL*, peripheral blood leucocytes; *HLH*, hemophagocytic lymphohistiocytosis; *bact. pneum.*, bacterial pneumonia; *CoV*, coronavirus other than SARS-CoV-2; *MIS-C*, multisystem inflammatory syndrome in children; *KD*, Kawasaki disease.

## Data Availability

This paper analyzes existing, publicly available data. The accession numbers for the datasets are listed in the key resources table. All original codes used for data analysis have been deposited at github (https://github.com/lschimke/COVID19-and-HLH-paper, accessed on 9 February 2022) and are publicly available as of the date of publication. R packages are listed in the key resources table. Any additional information required to reanalyze the data reported in this paper is available from the lead contact upon request.

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
