# Peer review of "Severe COVID-19 Shares a Common Neutrophil Activation Signature with Other Acute Inflammatory States"

_cells, 2022, doi:10.3390/cells11050847_

Round 1
Reviewer 1 Report
The study investigated the transcriptome of 1596 individuals, encompassing patients with COVID-19,
subjects with acute inflammatory states (HLH, multisystem inflammatory syndrome [MIS-C], Kawasakidisease, and different respiratory infections (seasonal coronavirus, 46 influenzas, bacterial pneumonia),
compared to healthy controls. Through a meta-analysis, the author provided evidence that severe COVID-19
shared activation of common cytokine pathways as well as dysfunction of neutrophil cell population with
other inflammatory states, expanding the knowledge on the systemic mechanisms involved in the
pathogenesis of severe COVID. Overall, the paper is well written and the hypothesis and purpose of the study are
clearly and concisely presented. In the materials and methods section, all research components are present and
clearly stated. the results are logically presented and in accordance with the significance of the findings.
Statements and conclusions are clearly supported by data and are linked to goals; however, study
limitations should be added. Tables and figures contribute substantially to content. References are
appropriate to manuscript type.
Author Response
Response to Reviewer 1 Comments
Point 1: Statements and conclusions are clearly supported by data and are linked to goals; however, study limitations should be added.
Response 1: We thank reviewer 1 for the critical analysis of our manuscript and acknowledge the positive response and recognition of the merits of our work. We included the study limitations in the discussion section. Please see line 538-546 of the revised manuscript.

Reviewer 2 Report
The life-threatening conditions imposed by COVID-19 arise from dysregulated inflammatory processes leading to acute respiratory distress syndrome (ARDS) and respiratory failure. However, anti-inflammatory therapies have not been successful and need refining. The current manuscript expands on previous work from this group that utilized RNA-Seq data sets from nasopharyngeal swabs and peripheral blood leukocytes from patients with SARS-CoV-2 to determine differentially expressed genes responsible for the disparity in severe symptoms arising in older and male patients (reference 21, https://doi.org/10.1172/jci.insight.147535); key neutrophil-related genes were identified.
Here the authors utilized gene expression data from publicly available sources to
- compare hyperinflammatory states in COVID-19 with other hyperinflammatory conditions
- compare gene expression signatures of COVID-19 patients in ICU with non-ICU
- define 11 critical genes that are dysregulated in patients with severe COVID-19
The bioinformatics in this study is extensive, the manuscript is well written, and the data are timely, considering the scourge that COVID-19 still presents. Particularly provocative is their identification of 11 specific genes in a neutrophil activation signature that could be beneficial in developing therapeutic interventions to suppress COVID-19 associated ARDS.
I have no major comments.
Minor issues for the authors to address include:
Throughout the manuscript. the authors could improve whether they're referring to a gene or a protein.
Line 126: …and COVID-19 published until February (missing the year. Probably 2022.)
Line 142: …and adjusted p-value < 0. (missing the cut off number)
Line 552-554 has problems. “Meanwhile other proteases (AZU1 and CTSG) and the inhibition of NET formation have been suggested to alleviate SARS-CoV-2 symptoms.”
Azurocidin (gene symbol AZU1) and cathepsin G (gene symbol CTSG) are proteins that are elevated in nasal swaps of COVID-19 positive patients (doi.org/10.1371/journal.pone.0240012), and are also members of the neutrophil activation signature identified in this study.
- Clarify if you’re referring to the gene or the protein
- “AZU1 and CTSG… have been suggested to alleviate SARS-CoV-2 symptoms.” is not consistent with the findings of this paper or the findings of others. Do the authors mean antagonizing these proteins? As written, it suggests augmenting these protein levels will be beneficial.
- Azurocidin does not have protease activity, and therefore should not be referred to as a protease.
Line 606: Conflicts of Interest: …serves on the Scientific Advisory Board of (missing).
Table 1: Define PBMC and PBL
Supplemental Table 14 column B has problems with formatting. Also, it’s not clear what columns E and F refer to. Please improve.
Similar in Supplemental Table 12. It’s not clear what columns E and F refer to.
Author Response
Response to Reviewer 2 Comments
We thank the reviewer 2 for the overall positive analysis of our manuscript. We appreciate the comments made by reviewer 2, which really helped to improve our manuscript. We took in consideration all his/her suggestions and edited the manuscript accordingly.
Point 1: Throughout the manuscript. the authors could improve whether they're referring to a gene or a protein.
Response 1: We thank reviewer 2 for this important observation and have used the italic spelling when referring to genes and normal spelling when referring to molecules and proteins throughout the text.
Point 2: Line 126: …and COVID-19 published until February (missing the year. Probably 2022.)
Response 2: We added the year 2021, which was the date for the curation of datasets for our first analyses of common transcriptome signatures between HLH and COVID-19. Please see revised manuscript line 118-121: “We searched public functional genomics data repositories (Gene Expression Omnibus [36] and Array Express [37]) for human transcriptome data from patients with HLH and COVID-19 published until February 2021 for our first analyses of common transcriptome signatures between COVID-19 and HLH.”
Point 3: Line 142: …and adjusted p-value < 0. (missing the cut off number)
Response 3: We added the cut-off for the adjusted p-value < 0.05 (revised manuscript line 137).
Point 4: Line 552-554 has problems. “Meanwhile other proteases (AZU1 and CTSG) and the inhibition of NET formation have been suggested to alleviate SARS-CoV-2 symptoms.”
Azurocidin (gene symbol AZU1) and cathepsin G (gene symbol CTSG) are proteins that are elevated in nasal swaps of COVID-19 positive patients (doi.org/10.1371/journal.pone.0240012), and are also members of the neutrophil activation signature identified in this study.
- Clarify if you’re referring to the gene or the protein
- “AZU1 and CTSG… have been suggested to alleviate SARS-CoV-2 symptoms.” is not consistent with the findings of this paper or the findings of others. Do the authors mean antagonizing these proteins? As written, it suggests augmenting these protein levels will be beneficial.
- Azurocidin does not have protease activity, and therefore should not be referred to as a protease.
Response 4: We thank reviewer 2 for the critical analysis of this sentence and agree with his/her comments to the referred proteins AZU1 and CTSG. We changed the sentence accordingly and hope that this improved the clarity of this sentence. Please see the complete sentence in line 561-564 of the revised manuscript as follows: “In addition, targeting other neutrophil proteins like Azurocidin (AZU1) and cathepsin G (CTSG) that are elevated in nasopharyngeal swaps of COVID-19 patients, as well as the inhibition of NET formation have been suggested to alleviate SARS-CoV-2 symptoms [97,101,124].”
Point 5: Line 606: Conflicts of Interest: …serves on the Scientific Advisory Board of (missing).
Response 5: We added the missing part of the sentence as you can read now as follows: “JJC is a consultant for Thermo Fisher Scientific and serves on the Scientific Advisory Board of 908 Devices.” Please see line 615-616 of the revised manuscript.
Point 6: Table 1: Define PBMC and PBL
Response 6: We added the definition of PBMC (peripheral blood mononuclear cells) and PBL (peripheral blood lymphocytes) to the footnote of Table 1. Please see in the manuscript lines 112-115.
Point 7: Supplemental Table 14 column B has problems with formatting. Also, it’s not clear what columns E and F refer to. Please improve.
Response 7: We reformatted Supplementary Table 14 and deleted column E and F.
Point 8: Similar in Supplemental Table 12. It’s not clear what columns E and F refer to.
Response 8: Thank you very much for this observation. We deleted column E and F in Supplemental Table 12.
